# The Impacts of National Centralized Drug Procurement Policy on Drug Utilization and Drug Expenditures: The Case of Shenzhen, China

**DOI:** 10.3390/ijerph17249415

**Published:** 2020-12-15

**Authors:** Lei Chen, Ying Yang, Mi Luo, Borui Hu, Shicheng Yin, Zongfu Mao

**Affiliations:** 1School of Health Sciences, Wuhan University, 115# Donghu Road, Wuhan 430071, China; leichen04@whu.edu.cn (L.C.); yangying@whu.edu.cn (Y.Y.); 2012302430049@whu.edu.cn (M.L.); 2018203050038@whu.edu.cn (B.H.); yinshicheng@whu.edu.cn (S.Y.); 2Global Health Institute, Wuhan University, 115# Donghu Road, Wuhan 430071, China

**Keywords:** National Centralized Drug Procurement policy, “4 + 7”, volume-based procurement, drug utilization, drug expenditures

## Abstract

In 2019, the Chinese government implemented the first round of the National Centralized Drug Procurement (NCDP) pilot (so-called “4 + 7” policy) in mainland China, in which 25 drugs were included. We conducted this study to examine the impacts of NCDP policy on drug utilization and expenditures, and to clarify the main factors contributing to drug expenditure changes. This study used drug purchasing order data from the Centralized Drug Procurement Survey in Shenzhen 2019. Drugs related to the “4 + 7” policy were selected as study samples, including 23 “4 + 7” policy-related varieties and 15 basic alternative drugs. Driving factors for drug expenditures changes were analyzed using A.M. index system analysis (Addis A. & Magrini N.’ method). After the implementation of the NCDP policy, the volume of “4 + 7” policy-related varieties increased by 73.8%, among which winning products jumped by 1638.2% and non-winning products dropped by 70.8%; the expenditures of “4 + 7” policy-related varieties decreased by 36.9%. Structure effects (0.47) and price effects (0.78) negatively contributed to the increase in drug expenditures of “4 + 7” policy-related varieties, while volume effects (1.73) had positive influence. NCDP policy successfully decreased drug expenditures of “4 + 7” policy-related varieties with structure effects playing a leading role. However, total drug expenditures were not effectively controlled due to the increasing use of alternative drugs.

## 1. Introduction

Many countries worldwide are facing the challenge of ever-increasing pharmaceutical expenditures [1,2,3,4], and the global pharmaceutical market reached USD 955 billion in 2019 [5]. In China, the total pharmaceutical expenditures increased by 17 times from 1990 to 2009, with an annual growth of 15.2% [6]. Moreover, total pharmaceutical expenditures accounted for 40% of total health expenditure in 2012 [7], far higher than the average level of 17% in the Organization for Economic Cooperation and Development countries [8]. In order to curb the growth of total pharmaceutical expenditures, the Chinese government has carried out centralized drug procurement since 2000 [9].

Globally, centralized drug procurement successfully saves drug costs [10,11]. With the help of a pooled procurement service, nine Caribbean island countries reduced unit drug costs by more than 50% [12]. The pooled procurement system implemented in Delhi, India, was estimated to save nearly 30% of the annual drug bill of the government, and this system was praised by the World Health Organization [13]. Joint procurement process achieved an estimated savings of 8.9% of drug costs in Jordan [14]. In addition, centralized drug procurement has reduced drug expenditures in Greece [15], Columbia [16], and Brazil [17]. In China, only a few studies have examined the effects of centralized drug procurement and reported inconsistent results. Li et al. [18] found that centralized drug procurement lowered prices of national essential medicines in Ningxia, Chongqing, and Tianjin provinces. However, Wang et al.’s study [19] in Jiangxi Province reported the opposite finding; the centralized tender system was insufficient to lower wholesale prices of national essential medicines. Besides, He et al. [20] demonstrated that centralized drug procurement failed to either bring down drug expenditures or combat its growth. Overall, Chinese literature paid more attention to the impact of centralized drug procurement on drug prices rather than drug expenditure. Usually, centralized drug procurement was not the main research factor in most literature, but only part of a larger policy portfolio. Furthermore, few studies seemed to delve into what are the decisive factors affecting the functions of centralized drug procurement. Therefore, it is unclear so far as to why centralized drug procurement has not effectively reduced drug expenditures in China but has indeed succeeded in other countries. More research is needed to explore the key factors that determine the role of centralized drug procurement in the Chinese context.

In January 2019, the General Office of the State Council of the People’s Republic of China implemented the National Centralized Drug Procurement (NCDP) policy, aiming at cutting drug costs and improving the drug procurement mechanism. In the first round of the NCDP pilot, 11 cities were selected as pilot cities to carry out drug volume-based purchasing, including four municipalities (Beijing, Tianjin, Shanghai, and Chongqing) and seven sub-provincial cities (Shenyang, Dalian, Xiamen, Guangzhou, Shenzhen, Chengdu, and Xi’an) in mainland China. Therefore, the first round of NCDP pilot is also known as the “4 + 7” pilot or “4 + 7” policy. The highlight of the “4 + 7” policy lies on the implementation of “volume-based procurement”, which indicates that the tenderee (government representative) clarifies the procurement volume (60–70% of the total annual drug use of all public medical institutions in pilot cities) when conducting tendering, and the tenderers (pharmaceutical manufacturer) quote according to this specific volume. The significance is to build up the contractual relationship between price and volume. “4 + 7” policy successfully tendered 25 drug varieties, of which 23 were generic products and 2 were branded products. In April 2019, the pilot cities successively implemented the tendering results of the “4 + 7” policy.

The “4 + 7” policy directly resulted in an average price reduction of 52% and a maximum price reduction of 96% for 25 winning products. However, a growing body of literature has clarified that price reduction alone does not necessarily lead to a decrease in drug expenditures. This is the so-called “bypass effect” which is very common in pharmaceutical policies [21,22]; that is, the expenditure of the drugs with price cuts steadily decreased, but the use of drugs without price cuts substantially increased. In “4 + 7” policy, the implementation of “volume-based procurement” seemed to ensure the use of low-priced winning drugs, although the effects of the policy on drug utilization and expenditures were still unclear. Given the shortage of existing evidence, we conducted this exploratory study to evaluate the general effect of the “4 + 7” policy. We first described the changes in volume and expenditures of different drug categories, for example, “4 + 7” policy-related drugs and basic alternative drugs, winning and non-winning products, generic and branded products. Then we analyzed the major influencing factors of drug expenditure changes under “4 + 7” policy.

## 2. Materials and Methods

### 2.1. Data Sources

This study used data from the Centralized Drug Procurement Survey in Shenzhen 2019 (CDPS-SZ 2019). In China, the CDPS-SZ 2019 was organized and conducted by the Global Health Institute of Wuhan University between December 2019 and January 2020. The survey aimed to evaluate the effect of drug-related policies in Shenzhen, China, and collected monthly drug purchase order data between 2017 and 2019. In the CDPS-SZ 2019 database, each purchase order record included purchase date, generic name, dosage form, specification, pharmaceutical manufacturer, price per unit, purchase volume, purchase expenditures, etc. A general database containing 963,127 monthly aggregated purchase order records was established, involving 1079 drug varieties (by generic name), 346 medical institutions, 857 pharmaceutical manufacturers. The total purchase expenditures reached RMB 20.87 billion.

This study aimed to examine the general effects of the “4 + 7” policy. Thus, we included samples with the following criteria: (a) the medication covered by “4 + 7” policy-related varieties and their basic alternative varieties (Table A1); (b) the time period between January 2018 and December 2019; and (c) the medical institutions purchasing drugs were from Shenzhen, China. Finally, 32,720 purchase order records of 38 drug varieties (by generic name) were included in the analysis.

### 2.2. Drug Classification

The “4 + 7” policy-related drugs and their basic alternative varieties were included as study subjects. The “4 + 7” policy-related varieties were sorted into winning and non-winning products based on the bidding results of “4 + 7” policy. Winning products referred to products that won the tender in “4 + 7” policy, otherwise they were deemed to be non-winning products. Besides, in terms of the original intention of policy design, “4 + 7” policy was trying to promote the replacement of branded drugs with generic drugs, so as to archive the goal of reducing drug costs. Thus, this study also dichotomized our samples into branded and generic products.

### 2.3. Outcome Variables

This study assessed the effect of “4 + 7” policy on both expenditure and volume of policy-related drugs. Expenditure data were reported in Chinese yuan, i.e., RMB. Volume was measured using the Defined Daily Dose (DDD), a measurement developed by the World Health Organization (WHO) to compare drug consumption. DDD refers to the average maintenance dose per day for a drug used for its main indication in adults, and is regarded as the international standard for assessing drug consumption across countries [21]. In this study, the DDD value of each product was determined according to the Guidelines for ATC classification and DDD assignment 2020 [23]. For two drugs, i.e., flurbiprofen axetil injection and pemetrexed disodium for injection, there is no DDD value assigned, thus they were not included in this study.

### 2.4. Statistical Analysis

Descriptive statistics were used. We first described the volume and expenditures of “4 + 7” policy-related drugs in the same period before (April to December 2018) and after (April to December 2019) the implementation of “4 + 7” policy. Then, we created graphical displays of the monthly purchase volume and expenditure of each drug group in order to observe and describe patterns over time from January 2018 to December 2019.

The A.M. index system analysis method (Addis A. & Magrini N.’ method) was used to assess different factors explaining the change in drug expenditures [24]. This method decomposes the change of drug expenditures into three dimensions: price effects, volume effects, and structure effects. The AM index system can be expressed as follows:(1)$$E=\frac{\sum P_{1}V_{1}}{\sum P_{0}V_{0}}=P\times V\times S=\frac{\sum P_{1}V_{0}}{\sum P_{0}V_{0}}\times \frac{\sum V_{1}}{\sum V_{0}}\times \left(\frac{\sum P_{1}V_{1}/\sum V_{1}}{\sum P_{1}V_{0}/\sum V_{0}}\right)$$
where:

*E* = Expenditures index, refers to the variations of drug expenditures in period 1 compared with period 0;

*P* = Price effects, refers to the pure price change in period 1 compared with period 0;

*V* = Volume effects, refers to the pure volume change in period 1 compared with period 0;

*S* = Structure effects, refers to the change in the mean price per DDD from shifts toward more or less expensive drugs;

*P*_0_, *P*_1_ = price per DDDs of each product in period 0, 1;

*V*_0_, *V*_1_ = DDDs of each product in period 0, 1.

In this study, period 0 is the base period and was assigned to April to December 2018, period 1 is the reporting period and was assigned to April to December 2019. In this model:

If *P* > 1, price effects increase the expenditure index. If *P* = 1, price effects have no impact on the expenditure index. If *P* < 1, price effects decrease the expenditure index.

If *V* > 1, volume effects increase expenditure index. If *V* = 1, volume effects have no impact on expenditure index. If *V* < 1, quantity effects decrease expenditure index.

If *S* > 1, structure effects increase the expenditure index. If *S* = 1, structure effects have no impact on the expenditure index. If *S* < 1, structure effects decrease the expenditure index.

## 3. Results

### 3.1. General Information

A total of 38 drug varieties (by generic name) were included in this study, including 23 “4 + 7” policy-related drugs (72 products) and 15 basic alternative drugs (34 products). The total purchase volume was 311 million DDDs, and the total purchase expenditure was RMB 1347.5 million. Of those, the volume and expenditures of “4 + 7” policy-related drugs were 225.0 million DDDs and RMB 1046.7 million, respectively. The volume and expenditures of basic alternative drugs were 86.1 million DDDs and RMB 300.8 million, respectively.

### 3.2. Change Trends of Drug Volume and Expenditures

#### 3.2.1. Winning and Non-Winning Products

Figure 1 demonstrates the monthly trends of volume and expenditures for winning and non-winning products. After implementation of the “4 + 7” policy, notable downward trends were observed for the volume and expenditure of winning products, while upward trends were observed for non-winning products. Compared with April to December 2018, the volume of winning products increased by 1638.2% during April to December 2019 and the expenditures decreased by 368.8%. The volume and expenditures of non-winning products dropped by 70.8% and 65.8%, respectively (Table 1).

#### 3.2.2. “4 + 7” Policy-Related Drugs and Alternative Drugs

According to the monthly trends (Figure 2), the expenditure of “4 + 7” policy-related drugs decreased, while the expenditures of basic alternatives and the volume of “4 + 7” policy-related drugs and their basic alternative drugs increased. After the implementation of the “4 + 7” policy, the volume of “4 + 7” policy-related drugs increased by 73.8% during April to December 2019 when compared with April to December 2018, while their expenditures dropped by 36.9%. Moreover, the volume and expenditures of basic alternative varieties increased by 26.0% and 24.6%, respectively (Table 1).

#### 3.2.3. Generic and Branded Products

Table 2 presents the change of volume and expenditures of generic and branded products. For “4 + 7” policy-related varieties, the volume and expenditure of generic products increased by 318.0% and 42.8% during April to December 2019 when compared with April to December 2018, respectively; while the volume and expenditure of branded products decreased by 58.1% and 59.4%, respectively. In terms of basic alternative varieties, the volume and expenditure of generic products increased by 22.4% and 21.6%, respectively; and the volume and expenditure of branded products increased by 29.0% and 26.4%, respectively.

### 3.3. A.M. Index System Analysis

Table 3 presents the results of A.M. index system analysis for “4 + 7” policy-related varieties and basic alternative varieties. In terms of “4 + 7” policy-related varieties, structure effects (*S* = 0.47) and price effects (*P* = 0.78) contributed negatively to the increase in drug expenditures, while volume effects (*V* = 1.73) had a positive contribution. In addition, volume effects (*V* = 15.52) contributed to the increase in drug expenditures for winning products and structure effects (*S* = 1.38) contributed to the increase in drug expenditures for non-winning products. For basic alternative varieties, volume effects (*V* = 1.26) and structure effects (*S* = 1.02) positively contributed to the increase in drug expenditures, while price effects played negative roles (*P* = 0.97).

The results of A.M. index system analysis for generic and branded products are shown in Table 4. For “4 + 7” policy-related varieties, volume effects (*V* = 4.12) contributed to the increase in expenditure for generic products and structure effects (*S* = 1.19) had a positive contribution for branded products. For basic alternative varieties, the decomposition analysis results of generic products and branded products were basically consistent with the whole basic alternative varieties.

## 4. Discussion

This study examined the impacts of NCDP policy on drug utilization and drug expenditures. The results suggested that NCDP policy exerted little influence on basic alternative varieties, whose drug utilization and drug expenditures held a mild growth trend throughout the study period. However, “4 + 7” policy-related varieties were obviously affected. The total drug utilization of “4 + 7” policy-related varieties increased, of which winning products soared and non-winning products declined sharply. The total drug expenditures of “4 + 7” policy-related varieties decreased significantly. Structure effects and price effects had negative influence on drug expenditures, while volume effects had a positive influence.

Containing costs is one of the inherent requirements of drug procurement [25]. Our results showed that NCDP saved 36.9% of drug expenditures on “4 + 7” policy-related varieties. Previous studies reported distinct results regarding cost-saving effect of related drug policies in different countries. Milovanovic et al.’s study [26] regarding drug tendering policy in Serbia reported a saving of 17.2% of the drug costs in a teaching hospital. Chaudhury et al. [13] mentioned that the Delhi Drug Policy for Essential Medicines was estimated to save nearly 30% in the annual drugs bill for the Government of Delhi. Perez et al.’s study [16] reported that centralized purchasing strategy for hepatitis C drugs in Colombia saved 91.6% of the drug cost. These differences in the above findings might be related to the different national condition and policy contents. In the first round of Chinese NCDP policy, all public health institutions in the pilot cities were involved, and original drugs as well as generic drugs which had passed the consistency evaluation of quality and efficacy of generic drugs were selected to conduct tendering; 25 drug varieties were finally included and most of which were medications for chronic diseases. These should be considered when it comes to the understanding and interpretation of present findings.

This study explored the role of price effects, volume effects, and structure effects in the changes in drug expenditures. For “4 + 7” policy-related varieties, structure effects acted as the key role, which was consistent with previous conclusions [27,28,29]. Structure effects, also known as “mixed effects” in some literature, refer to the shift in drug consumption toward more or less expensive products [27,30]. Our results indicated that the negative influences of structure effects on “4 + 7” policy-related varieties were due to the large number of low-priced winning products replacing high-priced non-winning products. This substitution effect was caused by the implementation of volume-based procurement, because doctors had to give priority to prescribing winning products to guarantee that the consumption of winning products reached the level promised by the government during tendering, otherwise the doctors would be punished. Obviously, the essence of volume-based procurement lay in restricting doctors’ prescribing behaviors. Many studies suggested restricting prescribing behaviors and promoting rational drug use in order to effectively control drug expenditures [27,29,31]. Our results seemed to confirm this view again.

Initiatives to contain pharmaceutical expenditures can be divided into supply- and demand-side [21,32]. Supply-side initiatives focus on obtaining low prices [22] and demand-side initiatives pursue the rational and economical use of drugs [33]. The lack of demand-side measures makes many policies inefficient [22,33,34]. The former centralized drug procurement in China was also merely a single supply-side initiative, because it only determined the price of medicines through tendering, however there was no guarantee that the winning products would be purchased by hospitals [35]. On the contrary, NCDP policy has taken both types of measures at the same time. On the one hand, the NCDP’s rule combination of “volume-based procurement + the company with the lowest price win the tender + single source of supply” put companies into a prisoner’s dilemma. In order to win the tender and maintain market influence, companies had to drastically cut prices. On the other hand, NCDP policy adopted a number of positive and negative incentives to enforce volume-based procurement, namely, compelling doctors to use the low-priced winning products. Therefore, the key to NCDP’s reducing drug expenditures might be the adoption of demand-side measures. The success of NCDP policy seemed to reaffirm the necessity of simultaneously applying supply-side and demand-side measures, which was provided as an example by other countries [36,37,38].

In this study, “4 + 7” policy-related varieties were also divided into generic products and branded products. Our results demonstrated that NCDP policy brought about cheaper generic products replacing branded products in abundance and ultimately led to a drop in drug spending, in line with Haas et al. [39] and Aalto-Setala et al. [40]. The effect of generic substitution has been confirmed in previous studies and this method is encouraged worldwide [41,42]. However, some literature has reported that switching from branded drugs to generic drugs would give rise to undesirable consequences, such as reduced treatment efficacy and tolerability [43,44,45]. Additionally, generic substitution is highly controversial and is often viewed with suspicion by healthcare providers and patients, because generic drugs do not have to prove efficacy through large-scale clinical trials [46]. To promote the implementation of NCDP policy in an orderly and stable manner, it is recommended to continuously monitor the quality and adverse reactions of winning generic products [47], and formulate the corresponding disposal plan. Moreover, strengthening the training of doctors and the publicity of patients, would ensure their understanding the value of winning generic products, so as to improve the acceptance by doctors and the adherence of patients [48,49,50].

Although rules have been perfected in new rounds, NCDP policy still needs improvements. For one thing, in the long run, a sustainable centralized drug procurement project needs a solid legal basis, or it will be questioned by stakeholders [10,47,51]. For another, given the steep growth in purchase volume of winning products, the risk of clinical irrational drug use increased. Lastly, the implementation of NCDP policy resulted in significant price differences of the same product in adjacent areas, leading to the phenomenon of institutional inequality.

Several potential limitations should be mentioned regarding the present study. Firstly, this study only considered basic alternative varieties provided by the monitoring program, rather than all the drugs which could replace the “4 + 7” policy-related varieties. Secondly, confounding factors such as the potential impact of other reform policies in Shenzhen have not been controlled, so there may be some deviation in the results. Thirdly, caution should be exercised in generalizing the findings. Only one of the 11 pilot cities was included in the study, so the results should be carefully extrapolated. Despite these limitations, the present study quantitatively evaluated the general effect of Chinese NCDP policy, and is the first to explore the influencing factors of drug expenditure changes under NCDP policy. The research results might be valuable references for the follow-up policy promotion and improvement.

## 5. Conclusions

After the implementation of the “4 + 7” policy, the winning products significantly replaced non-winning products, indicating a notable direct policy effect. The “4 + 7” policy-related drugs increased in volume and decreased in expenditures, while both the volume and expenditure of alternative drugs increased. The price of all drug categories dropped after the “4 + 7” policy was introduced, which played a positive role in drug cost control. The reduction in volume of non-winning products and branded products also played a positive role in drug cost control. The irrational use structure of non-winning products, branded products in “4 + 7” policy-related drugs, and alternative drugs contributed to the growth of drug expenditures. In order to perfect the policy, a monitoring mechanism for drug use needs to be established, especially for non-winning products, branded products, and alternative drugs.

## Figures and Tables

**Figure 1 ijerph-17-09415-f001:**
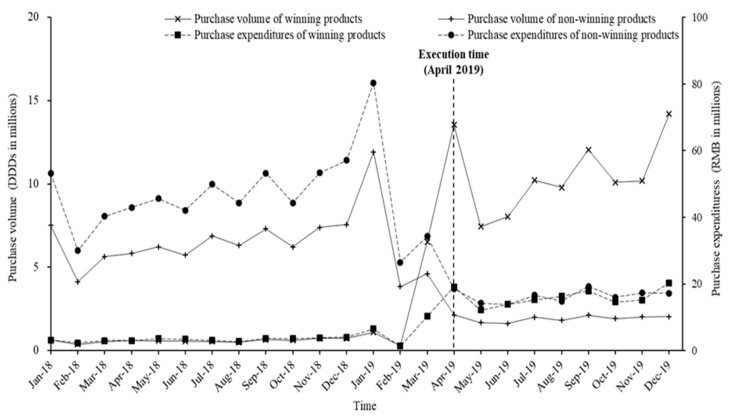
Monthly trends of volume and expenditures of winning and non-winning products among “4 + 7” policy-related drugs from January 2018 to December 2019.

**Figure 2 ijerph-17-09415-f002:**
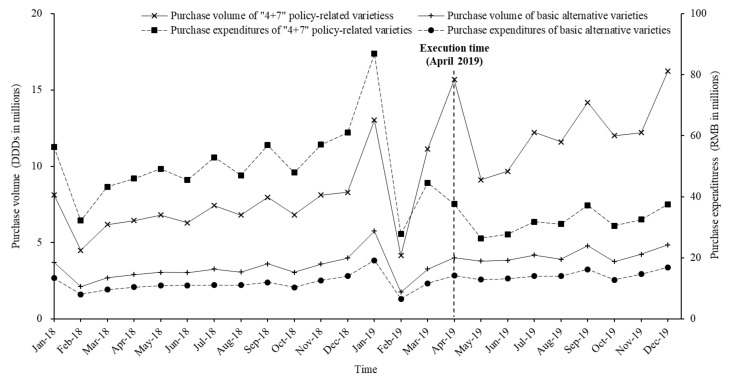
Monthly trends of volume and expenditures of “4 + 7” policy-related varieties and basic alternative varieties from January 2018 to December 2019.

**Table 1 ijerph-17-09415-t001:** Purchase volume, expenditures, and their year-on-year growth rate (YoY GR) of “4 + 7” policy-related varieties and basic alternative varieties in April–December 2018 and April–December 2019.

Drug Group	Volume (Million DDDs)			Expenditures (Million RMB)		
	2018	2019	YoY GR (%)	2018	2019	YoY GR (%)
“4 + 7” policy-related	65.0	113.0	73.8	463.4	292.2	−36.9
Winning	5.5	95.6	1638.2	30.8	144.4	368.8
Non-winning	59.5	17.4	−70.8	432.6	147.8	−65.8
Basic alternatives	29.6	37.3	26.0	103.5	128.9	24.6

**Table 2 ijerph-17-09415-t002:** Purchase volume, expenditures and their year-on-year growth rate (YoY CR) of generic and branded products from April–December 2018 and April–December 2019.

Drug Group	Volume (Million DDDs)			Expenditures (Million RMB)		
	2018	2019	YoY GR (%)	2018	2019	YoY GR (%)
“4 + 7” policy-related	65.0	113.0	73.8	463.4	292.2	−36.9
Generic	22.8	95.3	318.0	101.9	145.5	42.8
Branded	42.2	17.7	−58.1	361.5	146.7	−59.4
Basic alternatives	29.6	37.3	26.0	103.5	128.9	24.6
Generic	13.4	16.4	22.4	39.9	48.5	21.6
Branded	16.2	20.9	29.0	63.6	80.4	26.4

**Table 3 ijerph-17-09415-t003:** Results of decomposition analysis for “4 + 7” policy-related varieties and basic alternative varieties.

Drug Group	*E*	*P*	*V*	*S*
“4 + 7” policy-related	0.63	0.78	1.73	0.47
Winning	2.59	0.42	15.52	0.39
Non-winning	0.33	0.82	0.29	1.38
Basic alternative	1.25	0.97	1.26	1.02

*E*, expenditure index. *P*, price effects. *V*, volume effects. *S*, structure effects.

**Table 4 ijerph-17-09415-t004:** Results of decomposition analysis for generic products and branded products.

Drug Group	*E*	*P*	*V*	*S*
“4 + 7” policy-related	0.63	0.78	1.73	0.47
Generic	1.23	0.68	4.12	0.44
Branded	0.40	0.81	0.42	1.19
Basic alternative	1.25	0.97	1.26	1.02
Generic	1.22	0.98	1.23	1.02
Branded	1.26	0.97	1.28	1.03

*E*, expenditure index. *P*, price effects. *V*, volume effects. *S*, structure effects.

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
