# Peer review of "The Impacts of National Centralized Drug Procurement Policy on Drug Utilization and Drug Expenditures: The Case of Shenzhen, China"

_ijerph, 2020, doi:10.3390/ijerph17249415_

Round 1
Reviewer 1 Report
Authors must make the following corrections in the paper:
- Authors should explain the academic contribution of the work developed. Highlighting what is innovative / original about the existing literature.
- At the end of section 1 (Introduction), authors must present the structure of the paper.
- Authors should develop the conclusions of the work and refer in more detail to the next steps of the work
- Authors should add a new section to the paper: Methodology
- Authors should explain figure 2 better
Reviewer 2 Report
- Please perform a careful edit for syntax and grammatical errors.
- I would have liked to seen a more direct comparison to how the 4+7 study worked in other countries. I think a table comparing how China versus other countries performed would be beneficial. I also think a more detailed discussion about why some countries worked better or not would improve the discussion.
Reviewer 3 Report
This manuscript, entitled “The impacts of National Centralized Drug 3 Procurement policy on drug utilization and drug 4 expenditures: the case of Shenzhen, China” provides an overview of the effect of “ 4+7” policy on drug utilization and expenditures in mainland China. In general, the introduction provides enough background about the covered topic, the methods are adequately described, and the results are clear. Here, there are some comments that should be addressed before its publication.
- Line 33. It would be interesting to add more recent data (2019).
- Line 44: Greek should be Greece? Please, check this and correct if necessary.
- In all tables. I suggest including the line below tables (e.g 170) in table caption.
- Table A1: Have the authors planned to divide the drugs by therapeutic groups instead of including them all in one paragraph? It could be interesting.
- Line 226-227: I suggest extending this part of the discussion. It would be interesting to compare some of the factors that the authors mentioned.
Round 2
Reviewer 1 Report
The authors improved the paper. Thus, the paper can be published.